# The World Isn't Fair, but Shouldn't Elections Be? Evaluating Prospective Beliefs about the Fairness of Elections and Referenda

Jonathan Rose [1] and Cees van der Eijk [2,*]

1   Department of Politics, People, and Place, De Montfort University, Leicester LE1 9BH, UK; jonathan.rose@dmu.ac.uk
2   School of Politics and International Relations, University of Nottingham, Nottingham NG7 2RD, UK
*   Correspondence: cees.vandereijk@nottingham.ac.uk

**Abstract:** Almost all academic literature about the causes and consequences of fairness of elections and referenda is based on retrospective evaluations. One of the strongest findings of such studies is that nonvoting is higher among citizens who retrospectively perceived an election as unfair. However, on logical grounds, it is impossible to attribute lower rates of voting to retrospectively perceived unfairness because at the time of the vote citizens can only rely on their prospective expectations of fairness. Moreover, it is well documented that retrospective evaluations are strongly influenced by the outcome of the election which is, at the time of voting, still unknown. In view of the dearth of earlier studies on prospective views of electoral fairness, this article presents the first major exploratory analyses of determinants and consequences of prospective expectations of electoral fairness. Using data from Britain about expectations of fairness of three general elections and two referenda in the period between 2014 and 2019, it shows that the public hold mixed views about the fairness they expect to find when voting. The article demonstrates that these prospective fairness beliefs are sometimes noticeably different to retrospective beliefs in terms of their predictors. Moreover, in sharp contrast to literature based on retrospective evaluations, this article also finds that prospective evaluations do not importantly affect the decision to vote. These findings have important implications for how we understand and evaluate the inclusiveness of elections.

**Keywords:** electoral inclusiveness; electoral fairness; prospective fairness beliefs; winner–loser effects; Great Britain; Scottish independence referendum; Brexit referendum; general elections 2015; general elections 2017; general elections 2019; disability and fairness beliefs; electoral participation; turnout





## 1. Introduction

Voting plays a central role in contemporary democratic life. Elections are the standard means for determining who the leadership of a country will be and therefore also play a crucial role in setting the policy agenda of countries. Similarly, referenda can be a means of answering some of the most important and difficult policy questions facing countries. Because of the importance of elections and referenda, it is crucial that they are inclusive for all citizens. Inclusivity implies that the cost and effort required to participate in them are low and approximately equal for all citizens. This sometimes requires facilitation to aid citizens who would otherwise find it difficult to partake in the electoral process, such as disabled citizens who may face barriers to participation [1,2]. Inclusivity also presumes that electoral processes are conducted fairly, without manipulation and with accurate reporting of electoral results [3,4]. Moreover, an election may fail to be inclusive if it imposes differential psychological barriers on citizens–such as through differential beliefs about whether the election itself will be free and fair.

It is important that citizens believe that votes are fair even if the 'objective' level of fairness of an election is satisfactory. Such beliefs about electoral fairness exist in at least two different forms, each of which has important political consequences. The two forms

are, on the one hand, *prospective* expectations about the fairness of an election, and, on the other hand, *retrospective* perceptions of electoral fairness of an election. The latter of these has enjoyed considerable attention in social and political research. Their importance is often argued in terms of their link to beliefs about the legitimacy of the outcomes of the election in question, such that voters who think that an election was unfair are less likely to believe that it was legitimate [5–7]. Moreover, past literature has demonstrated that not only do processes that are perceived as fair increase perceptions of legitimacy, these perceptions of legitimacy in turn increase acceptance of decisions even in situations where those decisions are individually unfavourable (see, for example [8–10]). More recent studies have probed the interaction of perceptions of fairness with understandings of outcome favourability (see, for example [11,12]). In general, such studies argue that while beliefs about fairness have a clear role to play in explaining decision acceptance, the individual-level favourability of the outcome is both a significant predictor of acceptance and an important cause of retrospective perceptions of fairness [11] (p. 303). This literature is important for adding nuance to the debate about the role of fairness beliefs, but it does not undermine the importance of these beliefs; instead, it simply elaborates on the situations in which the effect of procedural fairness upon decision acceptance may be moderated (see also [13]). The sum total of this body of literature is an argument that beliefs that elections are perceived as unfair can undermine perceptions of the legitimacy of the outcomes which in turn can make citizens less likely to voluntarily comply with the policy results that flow from electoral outcomes and that this effect is particularly pronounced for those who did not get from elections what they had preferred. In essence, unfair processes risk losing the losers' consent [14].

The second form of fairness beliefs consists of *prospective* beliefs about how an upcoming election will be conducted. It is particularly these kinds of prospective beliefs that can be expected to affect how citizens engage with the electoral process and therefore affect its inclusivity. This engagement includes a variety of phenomena such as one's receptivity to the campaign in general and to the contributions of specific political parties, candidates, and others. It also includes one's electoral behaviour: whether or not one votes, and if so, for which party and candidate. From first principles, it is clear that these consequences can run in two different directions. Believing that the election will not be fair may for some voters diminish their incentive to turn out and vote (or to engage with it in other forms). But for others, it may work just in the opposite direction and mobilise them to have their voice heard anyway. Most likely, any effects of prospective fairness beliefs on electoral behaviour (participation and party choice) will be moderated by other attitudes and orientations such as political efficacy, sophistication, and partisan orientations. In any case, differences between groups in expectations of fairness are likely to result in differences in psychological, cognitive, and motivational barriers to engagement with the election, most likely undermining its inclusiveness.

The contrast in scholarly attention for retrospective and prospective electoral fairness is striking. Whereas retrospective survey questions have been included in many studies, including the comparative CSES round 1 that was fielded between 1996 and 2001 in 33 countries and territories, prospective expectations of fairness have been used relatively rarely [1]. As a result, much of the extant literature on consequences of electoral fairness beliefs is based on retrospective questions. Birch [15], for example, reports that evaluations of the fairness of elections are strongly interconnected with turnout in a range of elections, but her finding is based exclusively on retrospective evaluations of fairness. These findings are challenging to interpret given that a wide body of literature has established the existence of a 'winner–loser' effect following electoral competitions (see, for example [14,16–26]). This literature demonstrates that those who see their preferred side 'win' an election (usually understood in terms of seeing the party they voted for take charge of the executive) are more positive on a wide range of measures than those who see their preferred side 'lose'. While early literature similarly looked almost exclusively at retrospective evaluations—thereby finding that those who voted for the winning side were more positive on a wide

range of measures—this literature could not establish that there was a causal relationship between these variables. To address this limitation, some recent literature has instead made use of panel data to evaluate both prospective and retrospective evaluations to establish the individual-level changes during the electoral process (see, for example [25–27]), again finding (often strong) winner–loser effects. As such, we should strongly expect that the favourability of the outcome of the electoral process will cause substantive changes to beliefs about electoral fairness. Moreover, while this winner–loser literature often has very little to say about the effect on non-voters, who are frequently taken to count as missing data, where evaluations have been conducted, it appears that non-voters are more like losers attitudinally than they are like winners (see [28]). These considerations strongly suggest that post-election retrospective evaluations alone are unsafe for making claims about the impact of fairness beliefs on whether people do or do not turn out because these beliefs are in and of themselves heavily impacted by the very decision (i.e., to turn out or not) they are being used to explain. This leaves us with no previous literature upon which to base strong expectations about the impact of fairness beliefs on the decision to turn out.

Not only does the extant literature leave us in a weak position to form expectations about the effects of prospective beliefs about electoral fairness, but it also provides little information to allow us to understand the causes of the pre-existing beliefs about fairness. Previous literature has demonstrated that objective procedural fairness plays a crucial role in determining perceptions of the fairness of electoral processes [13]. Yet, in homogeneous national-level electoral settings, this factor is close to a constant for most citizens and as such is unable to account for the individual-level variance observed. Moreover, while it is apparent that outcome favourability can have an important role in conditioning retrospective perceptions of fairness [11] (p. 303), this cannot logically account for differences in prospective evaluations of the fairness of an election.

The research questions we address in this paper focus on prospective beliefs about the fairness of electoral competitions. We do this on the basis of a series of large (n ≈ 30,000) public opinion surveys collected in Great Britain between 2014 and 2019. In view of the dearth of literature on this topic, the analyses are inherently somewhat exploratory, though we use the literature concerning retrospective evaluations of fairness as a guide. The data cover five distinct electoral events: two referenda (Scottish independence referendum, 2014, and the EU membership or Brexit referendum, 2016) as well as three general elections (2015, 2017, 2019). Our specific research questions are: (1) How fair or unfair do British citizens expect these electoral events to be? (2) Which factors that have been suggested from studies of retrospective beliefs help explain prospective beliefs as well? (3) What are the consequences of prospective beliefs for electoral participation? Before we present our analyses, we first present a brief overview of the data and of the political context from which they derive. We then commence with a brief exploration of the responses to the prospective fairness question, covering matters of latent meaning, and of observed response distributions. We subsequently turn to analysing factors that may explain these beliefs. Finally, we investigate potential consequences of these beliefs, with a particular focus on electoral participation. In a concluding section, we reflect on our findings and their consequences for the inclusiveness of elections. In that section, we also discuss other potential consequences of beliefs about electoral fairness and the significance of these beliefs in the context of an advanced Western democracy.

## 2. Data and Context

In this study, we use data from the British Election Study Internet Panel (BESIP) [2]. This is one of very few large-scale representative surveys to include explicit questions about the perceived fairness of elections since the widely used Comparative Study of Electoral Systems (CSES) surveys from the late 1990s [3] (for a discussion of the CSES data collection, along with countries covered and perceived fairness values, see the Appendix of [14]; and [15]) [4]. BESIP is also important for including explicitly prospective questions about the expected fairness of elections, which allows for a true evaluation of the impacts of

fairness beliefs, unaffected by the winner–loser and outcome favourability effects discussed above. The prospective questions were asked as follows: 'How fairly do you expect the [relevant electoral event] to be conducted'. Each of these questions could be answered via a five-point rating scale with 1 labelled as 'conducted fairly' and 5 as 'conducted unfairly'. Respondents could also indicate 'don't know'. In this study, we focus specifically on waves which asked for prospective evaluations of fairness: Wave 2, Wave 7, Wave 11, and Wave 17. The total number of cases in each wave, along with the relevant electoral event, is shown in Table 1 below [5]. The final column of Table 1 shows that the questions about electoral fairness were sometimes presented to only a limited subsample of the respective waves. Whereas each wave contains between 25,000 and 30,000 respondents, the random subsamples who were asked these questions included only some 20% to 30% of the entire sample for Waves 2 and 17. As a result, the numbers of cases reported for our analyses below differ considerably between the five electoral events.

**Table 1.** British Election Study Internet Panel waves analysed.

| Wave Number | Dates Collected | Electoral Events That Fairness Question Relates to | Number of Respondents Presented with Prospective Fairness Question/Giving Valid Responses |
|---|---|---|---|
| W2 | 22 May 2014 to 25 June 2014 | Scottish independence referendum (18 September 2014) | 6047/5567 |
| | | 2015 UK general election (7 May 2015) | 5659/5193 |
| W7 | 14 April 2016 to 4 May 2016 | EU membership (Brexit) referendum (23 June 2016) | 27,526/27,060 |
| W11 | 24 April 2017 to 3 May 2017 | 2017 UK general election (8 June 2017) | 30,956/27,775 |
| W17 | 1 November 2019 to 12 November 2019 | 2019 UK general election (12 December 2019) | 8488/7212 |

Two of the prospective questions about expected fairness were asked in the same wave (Wave 2) but with respect to different electoral events and to different subgroups of respondents. One was asked to a random subset of the whole sample about expectations of the forthcoming 2015 general election and the other to Scottish respondents about the expected fairness of the Scottish independence referendum. While the 2015 general election was still a year away, and therefore the 'hot' phase of the campaign had not yet started, because of the Fixed-term Parliaments Act (2011), it was known exactly when the 2015 general election would be held, and parties were already preparing for that. In every other instance, the relevant questions were asked during the run-up to the electoral events in question, when campaigning was already actively happening.

BESIP is a long-term multi-wave panel study which aims to retain as many respondents as possible from one wave to the next. Nonetheless, we treat the five electoral events as independent replications and conduct our analyses predominantly in a form of separate cross-sectional studies. We therefore do not focus in this article on the evolution of fairness beliefs across multiple electoral events, mainly because 'simpler' questions about response patterns and potential drivers of the responses have to be addressed first. Of course, there exists overlap between the groups of respondents who have been asked the various electoral fairness questions, but that overlap is somewhat limited because of panel attrition and renewal, particularly over the five-year gap between the first data collection studied (Wave 2) here and the final (Wave 17). The overlap between groups that have been asked the electoral fairness questions is further reduced by these questions not always having been asked to every respondent in the sample but instead to random subsamples. Although BESIP can be considered as representative for the British adult population in many respects, its sample is in some respects biased. As is the case for virtually all survey-based election

studies [29,30], the sample under-represents non-voters. Averaged across the five electoral events, just 6.6% of BESIP respondents state they did not vote while (again, on average) 28.1% of the eligible population did not vote (based on data from [31]). This restricts to some extent analyses about the consequences of fairness beliefs for turnout, which are presented later. Nonetheless, particularly in the context of multivariate analyses, such over- or under-representations do not necessarily bias estimates of their relationships with other variables.

## 3. The Political Context of the Elections and Referenda

Of course, the elections and referenda probed here take place within a political context. The Scottish independence referendum of 2014 took place in a context where the Scottish government, the devolved executive for Scotland, was drawn from the pro-independence Scottish National Party who campaigned for independence. Between 2010 and 2015, the UK government was a Conservative–Liberal Democrat coalition led by David Cameron, which was against independence. While some polls suggested a tightening of public opinion near to the date of the referendum itself, most polling suggested that Scotland would remain part of the UK. The result following the referendum itself was a 55.25% win for 'No', a rejection of independence [32]. The 2015 UK general election had an incumbent Conservative–Liberal Democrat coalition government. The election would go on to deliver a Conservative majority allowing the Conservatives to govern alone, still under the leadership of Cameron. As a pre-election pledge, Cameron stated that he would hold an in–out referendum on the UK's membership of the EU—the so-called Brexit referendum. Cameron supported the pro-EU 'Remain' campaign and was significantly involved in campaigning. 'Remain' was also the official preferred option of the Labour party and was heavily supported by the Liberal Democrats who were the most pro-EU of the major parties. The anti-EU 'Leave' side was heavily championed by the United Kingdom Independence Party (UKIP), then led by Nigel Farage, but was also supported by a broad group of political actors from across the political spectrum. Notably, the Conservative politician Boris Johnson supported 'Leave', along with other members of David Cameron's cabinet including Michael Gove. This referendum was held on the 23 June 2016, resulting in a 51.9% vote for 'Leave' [33]. As a consequence of this result, David Cameron resigned as Prime Minister and was replaced in July 2016 by Theresa May. May was the incumbent Prime Minster, leading a Conservative majority government into the 2017 general election. This election was held as the government prepared to implement Brexit, though after the formal process of leaving the EU had been initiated. The 2017 general election was widely regarded as a misstep by May and forced the Conservative party to form a minority government relying on a 'confidence and supply' arrangement with the Northern Irish Democratic Unionist Party. May was effectively forced out of office in 2019 by members of her own party, partially in response to the withdrawal agreement she had negotiated with the EU. She was replaced by Boris Johnson, who became Prime Minister in July 2019. Because of leading a minority government, Johnson suffered a range of defeats to his legislative agenda, leading him to call the 2019 general election with his minority Conservative government as the incumbents. Johnson's Conservatives would win a large majority at the 2019 general election.

## 4. Responses to the Prospective Electoral Fairness Questions

Electoral impropriety has been rare in Great Britain in recent times. Detailed surveys of poll workers often find vanishingly few instances of even suspected fraud at the ballot box [34]. Of course, there have been occasional concerns about transgressions of electoral rules—often with respect to postal voting [35]—but rarely leading to sustained attention or, indeed, to evidence of violations of proper procedures on an appreciable scale [6]. Somewhat more attention has been paid to whether political parties always adhere to campaign spending limits and whether declarations of spending properly reflect the need to distinguish between national and local spending. While it is almost routine for the Electoral Commission to have to seek clarification on party reports, even where technical breaches are

found these are very rarely reported in the media and do not impinge upon broader public perceptions. Where breaches have been found to be more significant, these have attracted larger sanctions and have seen wider media coverage, such as when the Conservative party was fined GBP 70,000 for breaching campaign spending rules and filing incomplete records [36]. Nonetheless, this breach related to a small number of constituencies, and while it did generate media interest, this was not sustained. One could therefore wonder about the reactions of respondents when being asked about matters that have relatively rarely featured in public attention. How well-grounded are their responses and to what extent can these be regarded as informative of real opinions? We can consider this issue in several ways. The easiest, and perhaps most obvious, is to look at how many respondents are unable to answer the question about expectations of fairness in the upcoming electoral event. Table 1 reports the numbers of respondents who were asked the respective questions, as well as the number of them who provided a valid answer. From these numbers it follows that the percentage of respondents answering 'don't know' to the question probing expectations of electoral fairness was as follows for each of the five occasions:

- Scottish Independence Referendum 7.9% don't know
- 2015 General Election 8.2% don't know
- EU Membership (Brexit) Referendum 12.4% don't know
- 2017 General Election 10.3% don't know
- 2019 General Election 15.0% don't know

These percentages are not very high in the context of the percentage of 'don't know' responses for many other attitudinal questions in these same surveys [7]. Thus, the fact that issues on the probity of the electoral process did not figure very much in public discourse (particularly not in the electoral events prior to the EU referendum) did not present the samples of our surveys with insurmountable problems to express their opinion. The variation between these percentages is intriguing and invites speculation about what may drive greater (or lesser) degrees of certainty of expectations. Nonetheless, while this could profitably be considered in future research, we refrain from further analysis of the differences at this moment.

Another way to assess whether respondents' answers are reflecting a more or less cohesive understanding of the question is by considering the consistency of the answers given by respondents who were asked these questions more than once. This can be done with latent-variable analysis, for which purpose we use Mokken scale analysis [37,38] which is particularly applicable given the ordered-categorical nature of the five-point response scale [39] (p. 25). As the expected fairness questions were not always asked to all members of the sample, these analyses can only be performed for subsets of the total pool of respondents. The outcome of these analyses (which are reported in Appendix B) leads to the conclusion that the responses are not more or less random but that they are mainly driven by two substantive factors. The first of these is a weak 'generic expectation' factor that distinguishes respondents who—irrespective of the electoral event in question—have more optimistic or more pessimistic views of the fairness of elections and referenda [8]. The second is an event-specific factor that makes some of these referenda and elections generate more apprehensions about how fair they will be conducted than others, irrespective of where a respondent is located on the generic expectation factor. These apprehensions were smallest for the 2015 general election, highest for the EU referendum, and considerably higher for the 2017 and 2019 general elections than for the Scottish independence referendum.

The main conclusion to be drawn from these various analyses is that responses to the fairness questions we analyse here are not random or frivolous, but nor are they simple prejudices about the electoral system in general. Instead, they reflect genuine views of respondents that are importantly affected by the context during the run-up to the electoral events in question. This finding is in line with the experimental study by Doherty and Wolak [13] (p. 309), which finds that perceptions of fairness are importantly conditioned by (subjective evaluations of) the objective fairness of an event. As such, while it remains impossible to say exactly what (qualitative) meaning of fairness respondents had in mind

when answering, we can infer that their responses are grounded in a culturally shared colloquial understanding of what fairness of an electoral event is.

The obvious next question is how respondents answered the expected electoral fairness questions. Table 2 provides the distributions of the (valid) responses.

**Table 2.** Expectations of (un)fairness of different electoral events.

|  | Scottish Referendum | 2015 General Election | EU (Brexit) Referendum | 2017 General Election | 2019 General Election |
|---|---|---|---|---|---|
| (1) Will be conducted fairly | 41.2% | 38.2% | 28.7% | 32.2% | 27.6% |
| (2) | 17.8% | 23.6% | 17.4% | 18.1% | 23.0% |
| (3) | 18.1% | 20.7% | 23.0% | 22.8% | 23.7% |
| (4) | 13.1% | 10.5% | 16.8% | 14.8% | 17.1% |
| (5) Will be conducted unfairly | 9.9% | 7.1% | 14.1% | 12.1% | 8.6% |
| Number of valid responses | 5567 | 5193 | 27,060 | 27,775 | 7212 |

Table 2 shows that, in all instances, a plurality of respondents selected the most positive answer about how fair the election will be. Yet, in none of the occasions is this the majority of the answers. It is highest for the Scottish referendum, at around 41%, and lowest for the 2019 general election, where only 27.6% of respondents expected it to be fair without any qualification. In almost all instances (except for the EU referendum), we see a majority of responses for categories 1 and 2 of the response scale, which express an expectation of a fair or somewhat fair election or referendum. But even this softer criterion does not provide a ringing endorsement of expected fairness. For the EU membership referendum, these two categories collectively do not even reach a majority (46.1% fair or somewhat fair), and for both the 2017 and 2019 general elections, the 50% threshold is barely reached (50.3% and 50.5% fair or somewhat fair, respectively). Sizeable proportions of answers are in the somewhat non-committal middle category, which in four out of five instances is the second most common value of the distribution. Because of this category, the groups that expect the election or referendum in question to be conducted unfairly (or somewhat unfairly) are clear minorities (together ranging in size between 17.6% for the 2015 general election and 30.9% for the EU referendum). Yet, for an established political system that prides itself on its robust and resilient democratic character, these numbers should still be a cause for concern, particularly in light of older post-election studies that found very low levels of concern about the fairness of elections (see, for example [14], p. 145).

Obviously, these answers reflect subjective beliefs about fairness and are therefore grounded in personal expectations; they do not reflect anything like the unfairness that one observes in (quasi-)authoritarian states, such as explicit corruption, or politically motivated manipulation in counting and reporting election results, or partisan repression of voters [40]. Yet, it would, in our view, be unwise to discount the information in Table 2 as reflecting merely subjective beliefs without roots in objective fact. Respondents may well be motivated in their responses by their observation of matters that are perhaps not illegal yet could very well be considered as patently unfair; things such as pervasive use in campaigns of misinformation and untruths or attempts at character assassination [41,42]. They could even be motivated by flagrant violations of clear and legally binding rules, which may either not be discovered by authorities at all or may be discovered so long after the fact they were effectively unsanctioned for the conduct of the election in question [43].

After the analyses of response distributions reported in this section, we now turn to an analysis of factors that can be regarded as drivers of prospective fairness beliefs.

## 5. What Drives Expectations of Electoral Fairness or Unfairness?

Notwithstanding the broad interest that exists in understanding electoral integrity (see, for example [5,44–46]), there is surprisingly little evidence about the factors that condition

citizens' pre-election expectations of the fairness of these elections. Indeed, to the best of our knowledge, there are no earlier studies explicitly about such prospective expectations. The analyses to be reported in this section are therefore necessarily somewhat exploratory and will need to be confirmed in future research. However, relatively abundant literature exists which focuses on retrospective evaluations of electoral fairness in terms of the impact of being a winner or loser in the election (see, among many others, Refs [13,14,25,27,47,48]). This literature can be used to guide and inspire the subsequent analyses, even though, in our case, we are specifically interested in perceptions *before* respondents could know that they would be winners or losers. Doing so does not imply a firm expectation that correlates of retrospective perceptions of electoral fairness are necessarily the same as those of prospective expectations, but to the extent that they are different, they will help to better gauge the distinction between the two.

Not surprisingly, in view of their relevance for many other kinds of perceptions and orientations, a variety of social background factors have often been assessed for their associations with (retrospective) perceptions of electoral fairness. These include demographic factors such as gender, age, and ethnicity, important socialising factors such as religion and education, and various kinds of aspects of social status, including income and occupation. While these variables are not always significant and are not always consistent in directionality, common findings are that older people display more favourable perceptions of the fairness of elections, as well as members of socially dominant groups (in terms of ethnicity, religion, or language), those with higher social status, and those with higher levels of education (see, for example [49–51] (p. 225)). The effect of gender is more mixed, with some research finding men to be significantly more positive (for example [49]), some finding that men are significantly more positive in some elections and significantly more negative in others (for example [50]), and some finding insignificant effects (for example [45]). A second group of correlates of perceptions of electoral fairness consists of other perceptions of the 'wellbeing' of society, such as economic perceptions and perceptions of corruption, and the personal equivalents of such perceptions (whether one is in paid employment, financially secure, or, conversely, subject to adverse conditions) (see, e.g., [6,52] (p. 740), [53] (pp. 358–362), [54]). A third group of factors that have generally been found to correlate with perceptions of the fairness of elections are aspects of social and political integration such as social engagement, keeping abreast of social (and political) affairs, turning out to vote, and trust in social and political institutions [55]. A fourth and final group of characteristics that are commonly seen to be associated with perceptions of electoral fairness is political but of a more partisan kind: whether or not the political party one supports is an electoral winner [25,48] and ideological (left–right) orientation [49] (p. 313), [51] (p. 225), [54,56] (p. 85).

As already explained, the five electoral events for which the data contain (prospective) expectations about electoral fairness are analysed separately. A comparison of results from these five analyses provides a straightforward assessment of the consistency and stability of patterns of magnitudes and signs of associations between fairness expectations, on the one hand, and indicators of the various groups of potentially explanatory factors indicated above, on the other. In particular, we are interested in the directions and magnitudes of associations in the separate analyses; comparisons of the relative degrees of statistical significance are not useful given the large differences in numbers of responses for each of the five electoral events, as shown in Table 1. Nonetheless, given the absolute number of cases, in each analysis, variables that are not statistically significant are unlikely to have any substantively important effects upon prospective fairness evaluations.

The analyses proceeded as follows. The first stage consists of the identification in the BESIP data of suitable indicators for the kind of variables that have been used in the analysis of (retrospective) fairness perceptions, noted in the concise discussion above. In order to qualify as suitable, the variable in question must have been included in the BESIP questionnaires in waves shortly prior to each of the five electoral events under investigation. A second stage consists of investigating the bivariate associations of all selected variables in

order to avoid combinations of very highly collinear variables in further analyses and as a basis for combining information from different variables into a composite score, where this could validly be done [9]. A third stage involves a number of inductive regression analyses (for each of the five electoral events separately) to establish which of the potential drivers is related to fairness expectations in a consistent way. Consistency refers here to a combination of the direction and the magnitude of the regression coefficient; a variable is deemed to be consistent if it has the same sign and has a similar magnitude in at least three of the five regressions and is deemed to be inconsistent when the coefficients are non-negligible in magnitude and in different directions in two (versus three) out of five cases. Based on these analyses, the number of independent variables was gradually reduced until each of the remaining variables was consistent in the sense described [10]. The end result of this process consists of regressions of expected fairness (for each of the five electoral events separately) containing nine independent variables.

These regressions are displayed in Table 3, but before turning to them, it is important to report the variables that were found to be not, or not consistently (in terms of sign and magnitude), associated with fairness expectations: [11] gender; social grade; belonging (or not) to a religion; living as a single person or not; owning one's own home or not; ethnicity; being in paid work or not; whether or not one reads any daily newspaper; whether or not, if one reads a newspaper, it is a tabloid; whether or not one considers it a duty to vote; whether or not most people one knows do vote; and which party one intends to vote for at the next general election.

**Table 3.** Regressing expectations of electoral (un)fairness for different electoral events [a].

| | Scottish Referendum | 2015 General Election | EU Referendum | 2017 General Election | 2019 General Election |
|---|---|---|---|---|---|
| Age | −0.01 ** | −0.00 | −0.00 | −0.01 ** | −0.01 ** |
| Level of education | −0.02 | −0.07 ** | −0.13 ** | 0.03 ** | −0.04 * |
| Being disabled | 0.09 | 0.26 ** | 0.14 ** | 0.09 ** | 0.10 # |
| Feeling at economic risk | 0.04 ** | 0.06 ** | 0.06 ** | 0.06 ** | 0.08 ** |
| Political attention | −0.01 | 0.01 | 0.07 ** | 0.05 ** | 0.04 ** |
| Finding government and politics difficult to understand | 0.04 # | 0.04 # | 0.07 ** | 0.08 ** | 0.04 |
| Strength of party identification | −0.15 ** | −0.12 ** | −0.09 ** | −0.09 ** | −0.17 ** |
| Approval of incumbent government [b] | −0.24 ** | −0.05 * | −0.08 ** | −0.50 ** | −0.22 ** |
| Trust in MPs in general | −0.08 ** | −0.11 ** | −0.19 ** | −0.05 ** | −0.09 ** |
| Number of observations | 2867 | 2750 | 12,527 | 12,738 | 2789 |
| $R^2$ | 0.06 | 0.07 | 0.11 | 0.27 | 0.10 |

**: $p < 0.01$; *: $p < 0.05$; #: $p < 0.10$. [a] The dependent variable is coded such that '1' reflects an expectation that the election will be conducted fairly, and '5' reflects an expectation that the election will be conducted unfairly. All other operationalisations of variables are reported in Appendix A. [b] For the Scottish referendum, this is Approval of the Scottish Government; for all other electoral events, it is Approval of the UK Government.

Table 3 reports, for each of the five electoral events, a regression with nine independent variables empirically demonstrated to be consistently related to prospective evaluations of electoral fairness [12]. The full regression models, including also those variables that are inconsistently related to expectations of fairness, are reported in Appendix C.

The regressions lead to several important insights, particularly when used in conjunction with the prior analyses which revealed potential independent variables that are not, or not consistently, related to fairness expectations.

A first observation on the basis of Table 3 is that the explanatory power of the same set of independent variables (expressed in terms of $R^2$) differs strongly between the five electoral events, ranging from a low of 0.07 to a high of 0.27. This suggests that fairness expectations are to a considerable extent manifestations of short-term forces. This suggestion is strengthened by the observation that many stable background characteristics of respondents are not (or not consistently qua sign or magnitude) related to fairness expectations (i.e., gender; social grade; belonging (or not) to a religion; living as a single person or not; owning one's own home or not; ethnicity), while the magnitude of the coefficients of such stable variables that are consistently associated with fairness expectations can vary considerably from one electoral event to another (i.e., level of education, being disabled).

Table 3 displays a mixture of expected and unexpected coefficients. Older respondents have a small but consistent tendency to have higher expectations of electoral fairness than younger ones, and more highly educated citizens also display generally more optimistic expectations than less educated ones [13]. These patterns have also been reported in the literature (albeit with respect to retrospective perceptions of fairness) and are therefore not surprising. An important finding, particularly in view of the near-total neglect of this variable in the extant literature on electoral behaviour and related attitudes and orientations, is that being disabled is consistently associated with lower expectations of electoral fairness [14]. Also, in accordance with the extant literature is that respondents who feel personally at economic risk have lower expectations of fairness. As far as aspects of political involvement are concerned, it is not surprising that those who find government and politics difficult to understand have lower expectations of fairness, but it is surprising that this also holds for those who score higher in terms of political attention. Consistently strong effects on fairness expectations come from three political variables: strength of party identification (where absence of party identification is scored as the lowest level of strength), trust in MPs, and approval of the incumbent government. The first two of these are not partisan in character (i.e., they do not discriminate between supporters of various parties) but are more generic indicators of positive orientations to the existing political system. It is therefore not surprising (and, indeed, in accordance with other studies that focus on retrospective perceptions of electoral fairness) to find that respondents harbouring such positive orientations also have higher expectations of electoral fairness. The third of these three explicitly political variables is, although implicitly, partisan in character, as the government of the day will be of a certain political colour that is not to everyone's taste. Approving of the government is associated with higher expectations of fairness (and vice versa). This does not reflect mere party preference, however, as the inclusion or exclusion of the government approval variable in the more extensive regressions (reported in Appendix C) has either no substantive impact or at most only minor consequences for the effects of party preferences dummies (and vice versa); see also endnote 12 for these checks on the robustness of our findings.

It is also useful to consider the differences between the predictors of prospective and retrospective fairness beliefs directly. This is analysed in Table 4, which shows the relationship between the set of variables included in Table 3, along with party choice, for both prospective and retrospective evaluations of fairness. Party choice was not previously found to be consistently related to prospective fairness, which can also be seen with the inconsistency (qua sign and magnitude) of the pattern of effects of vote-intention dummies in Table 4 but is added here to allow for a consideration of winner–loser effects directly. There are several important findings from this analysis. Firstly, the predictors of prospective and retrospective fairness beliefs are frequently not consistent with each other for evaluations of the same electoral event. For example, level of education has a significant negative effect on prospective beliefs about the fairness of the EU referendum but a significant and positive effect on retrospective beliefs. Similarly, the effect of approval of the incumbent government switches direction from prospective to retrospective evaluations of the Scottish independence referendum. In the case of the 2017 general election, approval of the incumbent government retains the same direction, but the coefficient halves in size;

and similar effects can also be seen for strength of party identification. This should lead to caution about assuming that models that apply to retrospective evaluations of fairness should also apply without qualification to prospective evaluations. Secondly, the effects of party choice are not only inconsistent with regard to different electoral events (as noted above) but are also often inconsistent between prospective and retrospective evaluations of fairness of the same event. This is in part a reflection of winner–loser effects (see, for example [14]). The findings here serve as a caution against measuring winner–loser effects at only one point in time following an election. Moreover, the findings should stand as a caution about assuming who will feel like a winner based solely on who formed the government. Ahead of the 2017 general election, Conservative party supporters were significantly more likely than Labour party supporters to think the election would be fair, while after the election, there was no difference between them in average fairness beliefs, despite the fact that the Conservatives formed the executive following the election and the Labour party had no representation in government. Thirdly, the explained variance, measured via the R-square, of the models varies significantly both for the same models applied to different electoral competitions and for the same model applied to prospective and retrospective evaluations of the same electoral event. For prospective evaluations, the R-squares vary between 0.08 and 0.3, while the gap in R-square between prospective and retrospective evaluations of the 2017 general election is 0.2. Moreover, it is inconsistent whether models explain more variance for prospective fairness evaluations (as in the EU referendum and the 2017 general election) or retrospective fairness evaluations (as in the Scottish independence referendum and the 2019 general election). The sum total of these findings demonstrates that prospective evaluations of fairness are substantively different to retrospective evaluations, and future research needs to be mindful of this.

**Table 4.** A comparison of regression models predicting both prospective and retrospective evaluations of (un)fairness.

| | Scottish Referendum | EU Referendum | 2017 General Election | 2019 General Election |
|---|---|---|---|---|
| Age | −0.01 ** | −0.00 ** | −0.01 ** | −0.01 ** |
| | −0.01 ** | 0.00 * | −0.00 ** | −0.01 ** |
| Level of education | −0.01 | −0.08 ** | 0.01 | −0.04 * |
| | −0.04 # | 0.09 ** | −0.04 ** | −0.02 |
| Being disabled | 0.11 # | 0.12 ** | 0.09 ** | 0.09 # |
| | 0.27 ** | −0.02 | 0.11 ** | 0.01 |
| Feeling at economic risk | 0.04 ** | 0.06 ** | 0.05 ** | 0.07 ** |
| | 0.07 ** | 0.02 * | 0.04 ** | 0.06 ** |
| Political attention | −0.01 | 0.06 ** | 0.05 ** | 0.04 ** |
| | 0.00 | 0.04 ** | 0.04 ** | 0.05 ** |
| Finding government and politics difficult to understand | 0.05 # | 0.06 ** | 0.08 ** | 0.02 |
| | 0.02 | −0.01 | 0.08 ** | 0.00 |
| Strength of party identification | −0.13 ** | −0.07 ** | −0.08 ** | −0.18 ** |
| | −0.30 ** | −0.03 | −0.13 ** | −0.10 ** |
| Approval of incumbent government [a] | −0.23 ** | −0.17 ** | −0.43 ** | −0.20 ** |
| | 0.22 ** | −0.16 ** | −0.21 ** | −0.24 ** |
| Trust in MPs in general | −0.10 ** | −0.19 ** | −0.08 ** | −0.08 ** |
| | −0.17 ** | −0.01 | −0.10 ** | 0.00 |
| Party choice Conservative | 0.39 ** | 0.52 ** | −0.17 ** | −0.23 ** |
| | −0.14 | −0.36 ** | −0.02 | −0.40 ** |
| Party choice Labour | 0.11 | 0.02 | 0.08 * | 0.07 |
| | −0.16 # | −0.06 | −0.02 | 0.53 ** |

**Table 4.** *Cont.*

|  | Scottish Referendum | EU Referendum | 2017 General Election | 2019 General Election |
|---|---|---|---|---|
| Party choice Liberal Democrat | −0.12 | −0.01 | 0.05 | −0.05 |
|  | −0.19 | 0.07 | −0.09 # | 0.36 ** |
| Party choice SNP | 0.10 | 0.26 ** | 0.30 ** | 0.13 |
|  | 0.47 ** | 0.11 | 0.21 ** | 0.27 ** |
| Party choice UKIP (Brexit Party for 2019 general election) | 0.12 | 0.98 ** | −0.24 ** | 0.10 |
|  | 0.03 | −0.63 ** | −0.04 | −0.45 ** |
| Number of observations | 2368 | 12,006 | 12,789 | 2790 |
|  | 2763 | 10,002 | 10,600 | 8748 |
| $R^2$ | 0.08 | 0.18 | 0.30 | 0.12 |
|  | 0.23 | 0.07 | 0.10 | 0.20 |

Note: For each independent variable, the first coefficient pertains to a regression of the prospective fairness belief and the second to a regression of the corresponding retrospective fairness belief. For the 2015 general election, no retrospective fairness beliefs were asked, and hence it is not included in the table. OLS regression with listwise deletion of missing data, showing unstandardised coefficients. **: $p < 0.01$; *: $p < 0.05$; #: $p < 0.10$. The dependent variable is coded such that '1' reflects an expectation that the election will be conducted fairly, and '5' reflects an expectation that the election will be conducted unfairly. All other operationalisations of variables are reported in Appendix A. [a] For the Scottish referendum, this is Approval of the Scottish Government; for all other electoral events, it is Approval of the UK Government.

## 6. Consequences of Expected Fairness

The results so far demonstrate that prospective expectations of the fairness of elections are coherent perceptions that are related to a number of political and demographic variables. Nonetheless, the analysis also shows that the predictors of prospective expectations of fairness are not necessarily the same as the predictors of retrospective evaluations of fairness. One of the most important distinctions, of course, is that retrospective evaluations are inherently conditioned by the outcome of the electoral competition, whereas prospective evaluations cannot be. From our findings, the distinction survives the inclusion of variables concerning (prospective) party choice, which implies that the winner–loser effects are not foreshadowed in prospective expectations about outcomes. Given this, it is reasonable to question other distinctions that may exist between prospective and retrospective evaluations, not only in the predictors of fairness beliefs but also in the consequences of these beliefs. As noted above, retrospective evaluations of fairness have repeatedly been found to be very strongly related to electoral turnout [15,53,57,58]. Yet, given the distinctions between prospective and retrospective evaluations noted, and the strong and consistent findings of winner–loser effects conditioning a wide variety of political beliefs, it is reasonable to question the soundness of the interpretation that beliefs about fairness bring about differential electoral participation; particularly given the lack of testing of this relationship with prospective evaluations of fairness.

Because the BESIP data used here are a panel study, we are able to match up the prospective beliefs about fairness (expressed at some time before an election or referendum) with reported turnout in a later, post-election data collection. As such, we can evaluate the extent to which prospective beliefs impact upon the decision to turn out to vote in a way that avoids the confounding effect upon fairness beliefs of whether or not one voted for the winning party/parties. Because of the temporal ordering of the variables analysed (where prospective evaluations are recorded before the election and turn out decisions immediately after the election), the analysis is in principle very simple. Nonetheless, it is important to note that the level of electoral turnout reported by the BESIP samples is far higher than the true level of turnout in the elections or referenda that we look at. Averaged across the five electoral events, just 6.6% of BESIP respondents state not to have voted while (again, on average) 28.1% of the eligible population did not vote (based on data from [31]). This means that the absolute number of non-voters in the data collection is sometimes very

small, which naturally makes statistically detecting small effects challenging. Moreover, this inflated proportion of voters (versus non-voters) may render the samples, in this respect, somewhat less representative of the population as a whole [15]. Notwithstanding these obvious drawbacks of the BESIP data for analysing electoral participation, it is not possible for us to do anything more than acknowledge them and note that our findings should be seen as a starting point for further work on this topic.

We analysed the relationship between prospective fairness beliefs and electoral participation in several ways. A first straightforward analysis consists of inspecting bivariate crosstabulations between the two variables, without any controls. The results of this analysis are reported in Table 5, below. Although a chi-square criterion showed a significant relationship in four out of five instances (the only exception being the Scottish referendum), significance was particularly driven by relatively large numbers of observations [16]. Any linear or monotonic relationship is exceedingly weak, and in all instances, the relationship showed signs of (weak) non-linearity or non-monotonicity but not in a consistent pattern across all five elections and referenda. The potential problem with this analysis is that relevant relationships could conceivably be masked by not having taken into account other variables relevant when accounting for differences in electoral participation.

**Table 5.** Non-voting and electoral fairness beliefs (percentages non-voting per response category of fairness beliefs).

| | Prospective Fairness Beliefs | | | | | | Statistics |
|---|---|---|---|---|---|---|---|
| | **(1) Fairly** | **(2)** | **(3)** | **(4)** | **(5) Unfairly** | **Entire Sample** | |
| Scottish Referendum | 0.9% | 0.8% | 1.3% | 1.3% | 1.7% | 1.1% (n = 4292) | Chi-square = 3.43 (df = 4), $p = 0.488$<br>Cramér's V = 0.03 |
| 2015 General Election | 3.7% | 5.1% | 8.2% | 6.6% | 8.0% | 5.6% (n = 3961) | Chi-square = 24.90 (df = 4), $p < 0.001$<br>Cramér's V = 0.08 |
| EU Referendum | 3.4% | 4.8% | 5.9% | 4.3% | 3.7% | 4.4% (n = 20,571) | Chi-square = 44.13 (df = 4), $p < 0.001$<br>Cramér's V = 0.05 |
| 2017 General Election | 5.4% | 7.2% | 10.0% | 5.6% | 6.4% | 6.9% (n = 21,870) | Chi-square = 108.51 (df = 4), $p < 0.001$<br>Cramér's V = 0.07 |
| 2019 General Election | 6.0% | 7.1% | 12.2% | 5.4% | 10.2% | 8.0% (n = 4889) | Chi-square = 46.80 (df = 4), $p < 0.001$<br>Cramér's V = 0.10 |

A second analysis therefore consists of a block-recursive logistic regression of electoral participation, which is reported in Table 6. The first block of predictors is a baseline explanatory model containing a variety of control variables that are often used to explain voting behaviour, [17] and the second block adds expected electoral fairness as a predictor to this baseline model. The effect of the expected fairness variable in this analysis does not reach statistical significance in any of the five instances (despite the relatively large numbers of observations). Not only this, in no case does the addition of expected fairness beliefs increase the percentage of correctly classified cases. Moreover, in many cases, the addition of expected fairness actually reduced the AIC and BIC measures of model fit for many of the models.

**Table 6.** Block-recursive logistic regressions of turnout on baseline model and prospective fairness beliefs.

| | n (% Non-Voters) | | pseudoR$^2$ | Coefficient of Fairness Beliefs (*p*-Value) | Percentage Correctly Classified Cases | AUC-ROC [a] | AIC [b] | BIC [b] |
|---|---|---|---|---|---|---|---|---|
| Scottish Referendum | 2829 | Baseline model [c] | 0.125 | n/a | 99.36% | 0.797 | 214.69 | 286.06 |
| | (0.64) | Baseline plus fairness beliefs | 0.136 | −0.28 (0.125) | 99.36% | 0.795 | 214.37 | 291.69 |
| 2015 General Election | 2435 | Baseline model [c] | 0.088 | n/a | 96.30% | 0.736 | 726.19 | 795.76 |
| | (3.70) | Baseline plus fairness beliefs | 0.091 | −0.14 (0.124) | 96.30% | 0.739 | 725.88 | 801.25 |
| EU (Brexit) Referendum | 11,741 | Baseline model [c] | 0.082 | n/a | 96.88% | 0.728 | 3015.90 | 3104.35 |
| | (3.12) | Baseline plus fairness beliefs | 0.082 | 0.04 (0.331) | 96.88% | 0.727 | 3016.95 | 3112.77 |
| 2017 General Election | 3070 | Baseline model [c] | 0.088 | n/a | 95.50% | 0.742 | 1050.46 | 1122.81 |
| | (4.50) | Baseline plus fairness beliefs | 0.089 | −0.04 (0.560) | 95.50% | 0.743 | 1052.12 | 1130.50 |
| 2019 General Election | 3042 | Baseline model [c] | 0.094 | n/a | 94.81% | 0.720 | 1149.28 | 1221.52 |
| | (5.19) | Baseline plus fairness beliefs | 0.096 | −0.10 (0.145) | 94.81% | 0.729 | 1149.17 | 1227.43 |

Note: [a] The area under the curve (AUC) of the ROC (receiver operating characteristic) ranges from 0 to 1. It can be used as a measure of predictive performance of a model: higher values reflect models with higher predictive capacity. [b] AIC and BIC are, respectively, the Akaike and the Bayesian information criteria, each reflecting model fit while taking into account the degrees of freedom used (they both penalise models of which the complexity is not returned in terms of fit). Their values are not informative as such but are useful to compare models based on the same set of observations, as is here the case. For each: lower values reflect better fitting models. [c] The baseline model contains the predictors: gender; age; social grade; level of education; being disabled; perceived personal economic risk; political attention; trust in MPs; finding politics and government difficult to understand; strength of party identification; and approval of the UK Government. The n is the same for baseline and extended models (by forcing the baseline model to be estimated only for the complete cases involved in the extended model). The '% of non-voters' pertains only to this n.

The analysis reported in Table 6 has as a drawback that it assumes monotonicity in the relationship between electoral participation and fairness beliefs, while the cross tabular analysis displayed some (weak) suggestions of non-monotonicity. A third analysis therefore aimed to identify possible heterogeneity among respondents in the relationship between electoral participation and expected fairness. This is done by interacting expected fairness with, in turn, each of the variables of the block-1 base model and assessing whether any of the resulting interaction terms are significant. For each of the five elections and referenda, this yields nine additional models [18]. In the great majority of instances, the effects of these interactions are not statistically significant. In the few instances where they are, they are very small in magnitude and not consistent across electoral events in terms of the variables included in the interaction or in the direction of the interaction effect. We see as the most plausible interpretation of these rare significant effects that they are non-replicable results of capitalisation on chance, not deserving of further attention.

Our analyses, of course, do not rule out forms of heterogeneity that were not included in our specification of interactions. But the combination of nonsignificant effects in the great majority of instances, very weak and substantively inconsistent effects for the few interactions that are significant, and relatively large numbers of observations suggest strongly that any effects of (prospective) fairness expectations on turning out to vote are too weak and too idiosyncratic to serve as a useful basis for the development of hypotheses, theories, or even for useful descriptive understanding.

As we discussed in the introduction, consequences of expected fairness may well extend to other forms of engagement with the electoral process than electoral participation [6]. Our data do not offer optimal opportunities to explore these, nor is this paper the place to do so. But, as we discuss below, it would be implausible that politically relevant consequences would not exist, even if they do not seem to include—at least for Britain—effects on turning out to vote.

## 7. Conclusions

The perspective that perceptions of the fairness of elections matter are widespread among social scientists. This view is often grounded in an assumption that people who regard elections as unfair will also see them as illegitimate and will therefore be much less likely to voluntarily comply with unfavourable decisions that stem from those elections [5,7]. In this, there is a fear that if perceptions of the fairness of elections become too hostile, then the democratic process itself may be undermined, including citizens refusing to participate in elections [15]. Yet, the literature that seems to justify the worst of these fears has focused almost exclusively on retrospective evaluations of the fairness of elections. Where prospective expectations of fairness have been collected, they tend to be used only as a baseline against which changes in fairness perceptions can be evaluated (see, for example [25,55]). To be sure, there is a utility in knowing how the electoral process changes perceptions of fairness, and where we see large winner–loser effects, this may indicate a politicisation of the legitimacy of an election which is surely a serious issue. Nonetheless, such changes must be understood in relation to the baseline. Small winner–loser effects may still be extremely problematic if the losers start from a very low baseline while winners start from a very high baseline.

This article contributes to the literature on perceived fairness by elaborating on the baseline level of fairness within an advanced established democracy, by considering what factors predict these prospective expectations about the fairness of elections and referenda, and by showing how these initial expectations may (or may not) affect decisions to turn out to vote. The results of this paper demonstrate that the focus on retrospective evaluations in the extant literature misses important distinctions between prospective and retrospective evaluations. Prospective beliefs are not the same thing as retrospective beliefs; they do not necessarily have the same predictors, and they have a radically different relationship with potential consequences, such as turnout. While past literature finds that retrospective evaluations of fairness are very strongly connected with turnout [15], we find no such effects.

Our results are important for highlighting two key facts: (1) the absolute level of fairness expected by British respondents in recent elections and referenda is of a level that should be considered worryingly low. On average (across all five occasions), we find over 10% of the electorate expressing expectations, without reservations, that the election will be conducted unfairly (see Table 2). When taking the two categories together that reflect expectations of unfairness, we find (averaged across the five electoral events studied here) 25% of the population expressing hostile views about the fairness of elections. For evaluations of a democratic system that are as important as this, these numbers should be a concern. Moreover, (2) we find statistically significant differences between various groups of people in terms of their expectations of unfairness. In view of the common and ubiquitous understanding of the term 'fair' as implying equal treatment and the absence of an unjust advantage, these systematic differences in expectations (see Table 3) imply the need for systematic differences in cognitive, emotive, and physical effort in order to engage with the electoral process on an even footing. Some groups are systematically more likely to have to overcome the burdens emanating from their expectations of unfair treatment.

This is particularly notable in terms of disability status, where disabled people are consistently more likely to expect elections to be unfair. This probably reflects the reality of electoral administration that is frequently not sufficiently inclusive for disabled people. In a study by James and Clark [34] of poll workers in England, 9% of poll stations reported at least one disabled person having trouble accessing the polling station, and 14% of poll

stations reported at least one instance in which disabled citizens had difficulty filling out the ballot paper. Moreover, while these numbers already reflect a notable lack of inclusivity for disabled people, the results may present something of a best-case scenario given that some disabled people who have had problems engaging with the democratic process in the past may have stopped voting. As far as the expectations of disabled citizens are concerned, addressing these issues requires at least a systematic appraisal of electoral procedures from the perspective of electoral ergonomics (see [59]).

Making elections more inclusive requires as a necessary condition that they are expected to be fairer than is currently the case in Britain. In this paper, we identified widespread apprehensions of unfairness and showed how these apprehensions differ systematically between various groups (see Table 3). Our collective efforts should be directed to now identify how these misgivings can be ameliorated.

Although the data used in our study pertain to Britain, we suggest that our findings have a wider relevance than only to this single country for at least four reasons. First, Britain does not stand out as an outlier or deviant case in comparative studies of winner–loser effects and citizens' beliefs about procedural fairness. Most such comparative studies cover Western, developed liberal democracies (see, for example [14,15,49]. We consider it therefore likely that our major findings in this study will also hold in other Western liberal democracies. Second, the logical flaw of using retrospective beliefs to help explain citizens' electoral behaviour when we know (from the 'winner–loser' literature) that retrospective beliefs are heavily tainted by the knowledge of the outcome of an election is equally relevant in countries other than Britain; and almost all studies of fairness beliefs in other countries are based on retrospective data. In other words, on logical grounds, we must conclude that many results from retrospective studies of fairness beliefs in other countries should also be considered 'unsafe'. Third, although we demonstrated only for Britain that this is not simply a potential problem but is an actual problem, it stands to reason that, in other countries, the pattern of relationships between fairness beliefs and their potential determinants will often also be notably different for prospective and retrospective beliefs, thus necessitating re-evaluations of inferences based on retrospective data. Fourth, we cannot think of any reason why the lack of confidence in electoral fairness by disabled people would be a particularly British phenomenon, given that elsewhere too this group has largely to fend for themselves to solve the many practical problems that their condition throws at them, including those involving taking part in the electoral process. The size of this group is larger than often assumed, yet in spite of that, it is largely overlooked in electoral analyses, not only in Britain but also in other countries with strong traditions of electoral research.

**Supplementary Materials:** The following supporting information can be downloaded at: https://www.mdpi.com/article/10.3390/soc12030085/s1.

**Author Contributions:** J.R. and C.v.d.E. both contributed equally to the research conceptualisation, design, and analysis. The text was written collaboratively. All authors have read and agreed to the published version of the manuscript.

**Funding:** This research received no external funding.

**Institutional Review Board Statement:** Ethical review and approval were waived for this study because it is exclusively based upon anonymised secondary data that are curated for public reuse.

**Informed Consent Statement:** Informed consent was collected from all participants as part of the original data collection by the British Election Study.

**Data Availability Statement:** The BES Internet Panel is available for secondary analysis at the UK Data Service (see https://beta.ukdataservice.ac.uk/datacatalogue/studies/study?id=8810, accessed on 25 February 2022). The syntax of the analyses reported in this article is available in the form of a Stata *.do file in the supplementary material associated with this article.

**Conflicts of Interest:** The authors declare no conflict of interest.

## Appendix A

**Table A1.** Description of variables used in the analyses.

| Variable as Referred to in Article | BESIP Question Formulation (if Applicable) | Coded or Recoded Values | BESIP Variable Names NB: where variable name ends as, e.g., W2, this is part of the proper name of the variable and indicates it is from Wave2; where the name ends as Wx, the 'x' is a wildcard that stands for, respectively, 2, 7, 11, and 17, indicating the waves from which these variables were used |
|---|---|---|---|
| Gender | 'Are you . . . ?' | 1 = Male<br>2 = Female | gender |
| Age | | Coded numerically (i.e., '18' is 18 years, '43' is 43 years, etc.) | ageWx |
| Ethnicity | 'To which of these groups do you consider you belong?' | Recoded:<br>1 -> 1 (White British)<br>Else -> 0 (15 other ethnic groups) | p_ethnicityWx |
| Belonging to religion | 'Do you regard yourself as belonging to any particular religion, and if so, to which of these do you belong?' | Recoded:<br>1 -> 0 (No do not regard myself as belonging to any particular religion)<br>2 to 15 -> 1 (16 other religions/denominations) | p_religionWx |
| Living as a single person | 'What is your current marital or relationship status?' | Recoded:<br>1, 2, 4 -> 0 (Married; in a civil partnership, living with a partner but neither married nor in a civil partnership)<br>3, 5, 6 -> 1 (Separated but still legally married or in a civil partnership; in a relationship but not living together; single; divorced) | p_maritalWx |
| Social grade | | Coded as:<br>1 = A<br>2 = B<br>3 = C1<br>4 = C2<br>5 = D<br>6 = E | p_socgradeWx |
| Level of education | 'What is the highest educational or work-related qualification you have?' | Recoded by BESIP into:<br>0 = No qualifications<br>1 = Below GCSE<br>2 = GCSE<br>3 = A-level<br>4 = Undergraduate<br>5 = Postgrad | p_edlevelWx |

**Table A1.** *Cont.*

| Variable as Referred to in Article | BESIP Question Formulation (if Applicable) | Coded or Recoded Values | BESIP Variable Names NB: where variable name ends as, e.g., W2, this is part of the proper name of the variable and indicates it is from Wave2; where the name ends as Wx, the 'x' is a wildcard that stands for, respectively, 2, 7, 11, and 17, indicating the waves from which these variables were used |
|---|---|---|---|
| Being in paid work | 'Which of these applies to you?' | Recoded 1, 2, 3 -> 1 (Working full-time—30 h or more per week; working part-time—8–29 h per week; working part-time—less than 8 h per week) 4 to 7 -> 0 (Full-time student; retired; unemployed; not working) | p_work_statWx |
| Owning own house | 'Do you own or rent the home in which you live?' | Recoded 1, 2, 3 -> 1 (Own outright; own with a mortgage; own or part own through shared ownership scheme) 4 to 8 -> 0 (Rent private; rent local authority; rent housing association; I live with my parents, family, or friends but pay some rent; I live rent-free with my parents, family, or friends) | p_housingWx |
| Disabled | 'Are your day-to-day activities limited because of a health problem or disability which has lasted, or is expected to last, at least 12 months?' | Recoded 1, 2 -> 1 (Yes, limited a lot; yes, limited a little) 3 -> 0 (No) | p_disabilityWx |
| Perceived personal economic risk | 'During the next 12 months, how likely or unlikely is it that . . . . . . There will be times when you don't have enough money to cover your day to day living costs — You will be out of a job and looking for work' | Coded: 1 = Very unlikely 2 = Fairly unlikely 3 = Neither likely nor unlikely 4 = Fairly likely 5 = Very likely The answers to the two items were combined (additively), as they strongly reflect the same single underlying phenomenon (coefficient of homogeneity H is around 0.50 in all waves) | riskPovertyWx riskUnemploymentWx |
| Reading newspaper | 'Which daily newspaper do you read most often?' | Recoded: 1 to 15 -> 1 (any of named newspapers or 'other' mentioned) 16 -> 0 (none) | p_paper_readWx |

**Table A1.** *Cont.*

| Variable as Referred to in Article | BESIP Question Formulation (if Applicable) | Coded or Recoded Values | BESIP Variable Names<br>NB: where variable name ends as, e.g., W2, this is part of the proper name of the variable and indicates it is from Wave2; where the name ends as Wx, the 'x' is a wildcard that stands for, respectively, 2, 7, 11, and 17, indicating the waves from which these variables were used |
|---|---|---|---|
| Reading tabloid | Derived from 'Reading newspaper' before recoding | Recoded<br>1 to 5 = 1 (The Express; The Daily Mail or Scottish Daily Mail; The Mirror or Daily Record; The Daily Star or Daily Star of Scotland; The Sun; The Western Mail)<br>Else = 0 (all other newspapers) | |
| Considering voting a duty | 'It is every citizen's duty to vote in an election' | Coded:<br>1 = Strongly Disagree<br>2 = Disagree<br>3 = Neither agree nor disagree<br>4 = Agree<br>5 = Strongly Agree | dutyToVote2Wx |
| Most people one knows do vote | 'Most people I know usually vote in general elections' | Coded:<br>1 = Strongly Disagree<br>2 = Disagree<br>3 = Neither agree nor disagree<br>4 = Agree<br>5 = Strongly Agree | socialPressureVoteWx |
| Vote intention next general election | 'And if there were a UK General Election tomorrow, which party would you vote for?' | For analyses recoded into dummies for Conservatives, Labour, Liberal Democrats, SNP, UKIP. The reference category in regression analyses consists of all other parties mentioned (Green, Plaid Cymru, BNP, Brexit Party, and 'don't know') | generalElectionVoteWx |
| Ideology (left–right) | 'In politics people sometimes talk of left and right. Where would you place yourself on the following scale?' | Coded:<br>0 = Left<br>.<br>.<br>10 = Right | leftRightWx |
| Political attention | 'How much attention do you generally pay to politics?' | Coded:<br>0 = Little<br>.<br>.<br>10 = Much | polAttentionWx |

**Table A1.** *Cont.*

| Variable as Referred to in Article | BESIP Question Formulation (if Applicable) | Coded or Recoded Values | BESIP Variable Names NB: where variable name ends as, e.g., W2, this is part of the proper name of the variable and indicates it is from Wave2; where the name ends as Wx, the 'x' is a wildcard that stands for, respectively, 2, 7, 11, and 17, indicating the waves from which these variables were used |
|---|---|---|---|
| Finding government and politics difficult to understand | 'It is often difficult for me to understand what is going on in government and politics' | Coded: 1 = Strongly Disagree 2 = Disagree 3 = Neither agree nor disagree 4 = Agree 5 = Strongly Agree | efficacyNotUnderstandWx |
| Strength of party identification | 'Would you call yourself very strong, fairly strong, or not very strong *[name of party]*?' | Coded: 1 = Very strong 2 = Fairly strong 3 = Not very strong 4 = Not (i.e., indicated in previous question not to feel closer to any of the parties than the others) | partyIdStrengthWx |
| Approval of incumbent government | 'Do you approve or disapprove of the job that each of the following are doing? The UK Government The Scottish Government' | Coded: 1 = Strongly Disapprove 2 = Disapprove 3 = Neither approve nor disapprove 4 = Approve 5 = Strongly Approve | approveUKGovtWx approveScotGovtWx |
| Trust in MPs | 'How much trust do you have in Members of Parliament in general?' | Coded: 1 = No trust . . 7 = A great deal of trust | trustMPsWx |
| Expectation of electoral fairness(prospective) | 'How fairly do you expect the Scottish referendum to be conducted? ''Thinking of the general election for the Westminster Parliament that will take place in May 2015, how fairly do you expect it to be conducted? ''How fairly do you expect the EU referendum to be conducted? ''Thinking of the General Election for the Westminster Parliament that will take place June 8, how fairly do you expect it to be conducted? ''Thinking of the General Election for the Westminster Parliament that will take place December 12, how fairly do you expect it to be conducted?'' Thinking of the general election for the Westminster Parliament that took place on December 12th2019, how fairly do you think it was conducted?' | 1 = Fairly 2 3 4 5 = Unfairly | expectGoodConductScotRefW2 expectGoodConductGeneralW2 expectGoodConductEURefW7 expectGoodConductGeneralW11 expectGoodConductGeneralW17 |

**Table A1.** *Cont.*

| Variable as Referred to in Article | BESIP Question Formulation (if Applicable) | Coded or Recoded Values | BESIP Variable Names<br>NB: where variable name ends as, e.g., W2, this is part of the proper name of the variable and indicates it is from Wave2; where the name ends as Wx, the 'x' is a wildcard that stands for, respectively, 2, 7, 11, and 17, indicating the waves from which these variables were used |
|---|---|---|---|
| Belief of electoral fairness(retrospective) | 'How fairly do you think the Scottish referendum was conducted?<br>''How fairly do you think the EU referendum was conducted?<br>''Thinking of the general election for the Westminster Parliament that took place on [date of election], how fairly do you think it was conducted?' | | goodConductScotRefW3<br>goodConductEURefW9<br>goodConductGeneralW13<br>expectGoodConductGeneralW19<br>(NB: please note that last of these variables is mislabelled in the BESIP data file, as the variable reflects a retrospective and not a prospective fairness belief) |
| Turnout behaviour | 'Many people don't vote in elections these days. Did you vote in the referendum on Scottish independence that was held on 18 September 2014?''<br>Thinking back to the last UK General Election on 7 May 2015, a lot of people didn't manage to vote. How about you – did you manage to vote in the General Election in 2015?<br>''Talking to people about the EU referendum on 23 June 2016, we have found that a lot of people didn't manage to vote. How about you-did you manage to vote in the EU referendum?<br>''Thinking back to the 2017 General Election on 8 June, a lot of people didn't manage to vote. Howabout you? Did you manage to vote in the General Election?<br>''Talking to people about the General Election on 12 December, we have found that a lot of people<br>didn't manage to vote. How about you? Did you manage to vote in the General Election?' | Coded:<br>1 = Yes<br>2 = No'No I was not eligible to vote' coded as missing | scotReferendumRetroW3<br>genElecTurnoutRetroW6<br>euRefTurnoutRetroW9<br>genElecTurnoutRetroW13<br>genElecTurnoutRetroW19 |

## Appendix B. Latent Variable Analysis of Prospective Fairness Questions

To assess to what extent the responses to the expected fairness questions derive from a cohesive understanding of the items, we subjected them to a latent variable analysis. We used the non-parametric item-response procedure known as Mokken scaling [37,38], which is particularly applicable given the ordered-categorical nature of the five-point response scale, in contrast to factor analytic procedures [39] (p. 25). As the expected fairness questions were not always asked to all members of the sample, these analyses can only be performed for subsets of the total pool of respondents, or, when analysing all five items together, with pairwise deletion of missing data. The results of this analysis are reported in Table A2.

**Table A2.** Mokken scale analysis of responses to expected electoral fairness items.

| Electoral Event | Mean Score (1 = Fair; 5 = Unfair) | Item Coefficient of Homogeneity $H_i$ | z-Value of $H_i$ ($H_0$: $H_i \leq 0$) |
|---|---|---|---|
| Scottish independence referendum | 2.33 | 0.23 | 11.76 |
| GE2015 | 2.25 | 0.30 | 14.76 |
| EU membership (Brexit) referendum | 2.70 | 0.26 | 31.00 |
| GE2017 | 2.56 | 0.24 | 23.68 |
| GE2019 | 2.56 | 0.36 | 19.12 |
| All five items | | 0.26 | 26.46 |

The analyses demonstrate that all five items are positively associated with each other, with a magnitude of scalability that is close to the level of what is usually referred to as a weak scale (the usual cut-off being an overall H of 0.3, compared to the 0.26 seen here). These associations are statistically significant to a very high degree (as is indicated by the z-values). Moreover, the response patterns display no violations of the model assumption of monotone homogeneity (not shown in Table A2).

## Appendix C

**Table A3.** Extension of Table 3 with variables not included in further explanatory analyses (unstandardised OLS coefficients).

| | Scottish Referendum | 2015 General Election | EU Referendum | 2017 General Election | 2019 General Election |
|---|---|---|---|---|---|
| Variables retained in Table 3 (main text) | | | | | |
| Age | −0.01 ** | −0.00 | −0.00 * | −0.00 ** | −0.01 ** |
| Level of education | −0.02 | −0.05 ** | −0.06 ** | 0.00 ** | −0.06 ** |
| Being disabled | 0.13 * | 0.22 ** | 0.10 ** | 0.09 ** | 0.10 # |
| Feeling at economic risk | 0.05 ** | 0.05 ** | 0.06 ** | 0.05 ** | 0.07 ** |
| Political attention | −0.02 | 0.01 | 0.07 ** | 0.04 ** | 0.04 * |
| Finding government and politics difficult to understand | 0.03 | 0.05 * | 0.06 ** | 0.08 ** | 0.02 |
| Strength of party identification | −0.14 ** | −0.15 ** | −0.05 ** | −0.07 ** | −0.18 ** |
| Approval of incumbent government [a] | −0.23 ** | −0.05 # | −0.18 ** | −0.43 ** | −0.20 ** |
| Trust in MPs in general | −0.11 ** | −0.09 ** | −0.19 ** | −0.08 ** | −0.07 ** |
| Variables not related or inconsistently related to dependent variable | | | | | |

**Table A3.** *Cont.*

| | Scottish Referendum | 2015 General Election | EU Referendum | 2017 General Election | 2019 General Election |
|---|---|---|---|---|---|
| Social grade | 0.00 | 0.00 | 0.02 * | −0.01 | −0.01 |
| Gender | −0.06 | 0.05 | −0.11 ** | −0.00 | −0.03 |
| Religion | −0.01 | 0.04 | 0.01 # | −0.00 | −0.01 |
| Being single | −0.01 | −0.08 | −0.01 | −0.00 | −0.00 |
| Homeowner | 0.09 | −0.22 ** | 0.00 | 0.00 | 0.07 |
| Ethnicity | −0.11 | −0.16 # | 0.03 | −0.29 ** | 0.05 |
| In paid work | 0.02 | 0.01 | 0.06 * | −0.00 | −0.00 |
| Does not read a newspaper | −0.08 | −0.05 | −0.03 | 0.00 | −0.07 |
| Good time to purchase | −0.04 # | −0.01 | −0.05 ** | −0.06 ** | −0.02 |
| Duty to vote | −0.00 | −0.05 # | −0.06 ** | n/a | −0.09 ** |
| Social pressure to vote | 0.00 | −0.02 | −0.04 * | n/a | n/a |
| Party choice dummies (5 largest parties only) | | | | | |
| Vote intention Con | 0.36 * | −0.14 | 0.56 ** | 0.16 ** | −0.25 ** |
| Vote intention Lab | 0.10 | −0.21 * | 0.02 | 0.08 * | 0.07 |
| Vote intention LibDem | −0.11 | −0.17 | 0.01 | 0.04 | −0.03 |
| Vote intention SNP | 0.06 | 0.00 | 0.28 ** | 0.34 ** | 0.12 |
| Vote intention UKIP | 0.04 | 0.01 | 0.97 ** | −0.22 ** | 0.11 |
| Number of observations | 2516 | 2355 | 9769 | 11,062 | 2438 |
| $R^2$ | 0.08 | 0.08 | 0.18 | 0.30 | 0.12 |

OLS regression with listwise deletion of missing data, showing unstandardised coefficients. **: $p < 0.01$; *: $p < 0.05$; #: $p < 0.10$. The dependent variable is coded such that '1' reflects an expectation that the election will be conducted fairly, and '5' reflects an expectation that the election will be conducted unfairly. All other operationalisations of variables are reported in Appendix A. [a] For the Scottish Referendum, this is Approval of the Scottish Government; for all other electoral events, it is Approval of the UK Government.

## Notes

[1]  Sometimes survey questions are explicitly prospective or retrospective by including a reference to a specific election. In other instances, they are retrospective or prospective only implicitly by the timing of the fieldwork: shortly before or shortly after an election (as is the case for the National Annenberg Election Study—see [27]). We suspect that implicit questions may be somewhat susceptible to contain elements of generic system support, while the explicit questions focus more on the fairness of concrete electoral events. In some instances, it will be impossible to determine whether a question is prospective, retrospective, or mainly system support, as in the case of the European Social Survey (round 6), which lacked a reference to any specific election; the British sample of this study was conducted in 2012, around halfway between the general elections of 2010 and 2015. A similar lack of reference to a specific election occurred in the Electoral Commission's survey of public attitudes that was conducted in January–February 2021, outside an actual election context, and therefore also likely to measure aspects of generic system support rather than specific prospective or retrospective electoral fairness beliefs (see [60]).

[2]  The BES Internet Panel [61] is available for secondary analysis at https://www.britishelectionstudy.com/data-objects/panel-study-data/ (accessed on 31 January 2022).

[3]  In between the CSES and the BESIP studies, we find in Europe the European Social Survey (round 6, 2012) which was fielded in 29 countries and asked about the extent to which elections in one's country were seen as 'free and fair' (but without a clear prospective or retrospective direction); and in the USA, the National Annenberg Election Study which included a slightly amended version of the CSES question in 2008, 2012, and 2016.

[4]  Note a potentially important difference between the CSES and BESIP questions: the CSES primed people to think about fairness of the last national election relative to standards in other countries. The question was asked as "In some countries, people believe their elections are conducted fairly. In other countries, people believe their elections are conducted unfairly. Thinking of the last general election in [country], where would you place it on this scale of one to five where ONE means that the last election

was conducted fairly and FIVE means that the last election was conducted unfairly?" (See, for example [62], which included the question).

5   BESIP also included explicitly retrospective questions about some of the same electoral events. Analyses have been conducted upon both the prospective and retrospective fairness questions in a difference-in-differences analysis, which showed very strong winner–loser effects for the EU membership (Brexit) referendum [25].

6   The Electoral Commission reviews procedural aspects of elections and the public's satisfaction with electoral processes. These reports uniformly found the general elections of 2015, 2017, and 2019 to have been procedurally well run (see [63–65]). Moreover, even allegations of electoral fraud are very rare. In 2019, when a wide range of elections were held, including the general election of 2019 studied here, across the whole country, only 595 cases of alleged electoral fraud were investigated by the police, only six of which led to a conviction or a police caution [66].

7   For example, a simple open-ended question about what one sees as the most important issues in the country yields around 10% 'don't knows'; questions about the Conservatives' and Labour's position with respect to redistribution of income give between 20 and 30% 'don't knows'; and questions about preferences on European integration (asked before the EU referendum) lead to more than 10% 'don't knows'.

8   This conclusion can be drawn from the fact that the various items about prospective fairness are significantly and positively associated in terms of scalability (even though scalability as a whole falls slightly below the standard cut-off of 0.3) and the fact that the items as a set do not violate the monotone homogeneity assumption of the Mokken model (see Appendix B).

9   Such composite scores are only constructed—using appropriate latent variable models—if the variables involved could be demonstrated to reflect a single underlying (latent) variable.

10  During this process, the coefficients of all variables were monitored continuously so as to avoid that this inductively driven process would inadvertently result in the elimination of very strongly associated variables which, even if not consistent, could be helpful in understanding the responses to the fairness expectation questions.

11  Appendix A provides details about the operationalisation of these variables.

12  The operationalisation of the variables is reported in Appendix A. The analyses reported are straightforward OLS regressions. In view of the categorical nature of the dependent and some of the independent variables, the regressions were also specified as ordered logit models, and categorical independent variables were also specified as dummies. These alternative specifications of the regressions give rise to the same substantive conclusions as OLS, which is reported for reasons of simplicity of presentation. Within the OLS regressions, none of the coefficients changes in sign or approximate magnitude if one of the other independent variables is deleted from the regression equation. The regression analyses reported in this article were conducted using listwise deletion of missing data, which accounts for the reduction in numbers of observations when compared to Table 1. In principle, this could be addressed by multiply-imputing the missing values, which would strongly increase the number of cases in the analysis. However, such a multiple imputation approach makes assumptions about the process(es) that generated the missing data that are impossible to test. If these assumptions are violated, multiple imputation can produce biased regression coefficients even in situations where a complete-cases approach (i.e., listwise deletion) would not produce bias [67] (p. 4). In a simulation study, multiple imputation has been found to be superior to a 'complete cases' approach in terms of bias to coefficients only where the data are missing at random conditional on the dependent variable [67] (p. 4). In this study, this would be the case if those who regarded elections as unfair were systematically less likely to respond to the questions we analyse (perhaps out of fear of reprisals). While this assumption may be reasonable in authoritarian systems, and as such, analysts should be mindful of this issue when replicating this study, it is implausible in the case of a Western liberal democracy with an enshrined right to political speech and where the data are collected in an anonymous format by people who are not affiliated with the government.

13  The only exception here is for the 2017 general election, where the coefficient was both significant and differently signed. This is a finding that cannot be explained by reverse coding or other data problems and therefore appears to represent a substantive difference at this election from the other electoral events studied here.

14  The importance of this finding is underlined by the magnitude of this group: consistently, across all waves of BESIP, approximately three in ten respondents indicate to suffer from some kind of disability. This is a higher proportion than that suggested by 2018 data from the Papworth Trust, which comes to 20% of the UK population [68], but is based on a precise legal definition that may not correspond with people's own classification of being disabled.

15  This difference with the general population is plausibly the effect of two factors. One is (self-)selection, as people who are very uninterested in and disengaged from politics are unlikely to volunteer for being a respondent in surveys that strongly focus on political matters. A second factor that distinguishes the samples from the general population is a learning or socialising effect: being surveyed repeatedly about political matters tends to increase one's interest in politics.

16  The smallest number of observations in these five analyses is 3961. The weakness of these associations—in spite of their significance—is easily illustrated by the case of the EU membership referendum: while n = 20,571, chi-square reaches only 43.19 (df = 4), or expressed in other terms, Cramer's V = 0.036 and pseudo-R-squared = 0.003.

17  This block contains the following variables: gender; age; social grade; level of education; being disabled; perceived personal economic risk; finding politics and government difficult to understand; strength of party identification; and approval of the UK Government.

18  Because these interaction models generate so many separate models, it is not possible (or particularly desirable) to reproduce them all in this article. Here, we focus on the substantive conclusions of all of these models collectively.

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
