# Peer review of "The World Isn’t Fair, but Shouldn’t Elections Be? Evaluating Prospective Beliefs about the Fairness of Elections and Referenda"

_societies, doi:10.3390/soc12030085_

Round 1

Reviewer 1 Report

This manuscript investigates prospective expectations/beliefs about election fairness in Britain, over three general elections and two referenda between 2014 and 2019.

Overall, I found the paper interesting and relevant, especially given claims made in recent years by several politicians in different countries regarding election fraud. The author(s) are also doing a good job motivating the paper (esp. pp. 1-2). That said, there are several aspects the authors need to clarify and revise before this paper is suitable for publication. I consider comments 1-2 as the more important comments, and the other comments are more minor.

1) First, the paper would benefit from the authors elaborating on how survey respondents (in the UK or elsewhere) seem to understand the concept of "electoral fairness". Specifically, how respondents understand or evaluate the dependent variable in this paper (i.e., "how fairly do you expect the [election] to be conducted?"). Given that "electoral impropriety has been rare in Great Britain in recent years", as the authors note on page 6, it is unclear what people have in mind when 20-25% of the public answers that it will not be conducted fairly (Table 2, p. 7). The authors themselves note that different respondents might have in mind different things when they consider a prospective election's fairness (e.g., p. 8, lines 314-320).

Such an elaboration, I believe, would help the readers appreciate the authors' results. Otherwise, it might be that different respondents have very different things in mind when they answer the dependent variable, making it difficult to understand whether the results attest to a serious democratic problem, or perhaps to some respondents just "blowing-off-steam" and denigrating the establishment.

2) The authors claim that the results of their analyses show that prospective beliefs about election fairness have different predictors than retrospective beliefs about election fairness, and, notably, that the former, unlike the latter, "are not consistently affected by party preferences" (from the abstract). However, my understanding is that the variables the authors use in the current manuscript are not the same as those used in studies of retrospective beliefs about election fairness. Namely, the current authors do not have a "winner (versus loser)" dummy variable as a predictor in their regression analyses (e.g., Table 3, page 11). Moreover, the variable most resembling of such a dummy variable, i.e., the "Approval of incumbent government" variable, is significant and has the expected sign in all five elections/referenda, which runs counter to the authors' argument of substantial differences between prospective and retrospective beliefs about election fairness.

Thus, I believe the authors should provide more evidence for their claim of said substantial differences between prospective and retrospective beliefs. They could do so by adding a dummy variable for reported voting for the ruling party (i.e., the "winner") in the last election when they analyze the three elections (not sure such an analysis would be relevant for the two referenda). Otherwise, they are risking confusing their readers.

Minor comments

3) I believe that the paper would benefit from the authors clearly stating in the abstract and the introduction that this paper provides an exploratory examination of prospective beliefs about election fairness, and not a "confirmatory" examination with hypotheses testing.  Reading the abstract, I was under the impression that this paper would test hypotheses; and reading the paper, which elaborates, inter alia, on the share of "don't know" responses, and later on predictors of prospective election fairness beliefs and their "effects", I was a bit confused. Nothing wrong with an exploratory examination; but this needs to be acknowledged.

4) Relatedly, it was not clear to me why did the authors present in Table 3 only the variables that they considered as "consistent" predictors of prospective election fairness beliefs. I mean, given that this is an exploratory paper, why not show all results?

5) Also relatedly, on pages 12-14 the authors provide analyses in which they examine the "consequences" of expected fairness. I did not find these analyses very convincing (and no regression table was provided), but, more importantly, given that this is an exploratory paper with exploratory analyses, I was not sure why the authors used a "causal language" instead of tentatively testing whether prospective election fairness beliefs is correlated with additional politically-relevant variables, such that future studies could more clearly test if prospective election fairness beliefs indeed causally affect these politically-relevant variables.

6) Can the authors explain, even if only in a paragraph, what researchers from other countries can learn from the British case study presented in this paper? I would appreciate it if the authors addressed potential issues of generalizability (or lack of) from this case study to other case studies/countries.

Author Response

Please see attchment

Reviewer 2 Report

I found the paper interesting, and it gives us some new insights to understand how beliefs about electoral fairness correlate with different political and socio-demographic factors. I like the authors' approach to using prospective fairness instead of the usual retrospective one. Although the results are interesting, I was not convinced by the authors' analysis. Here I list my major concerns with the paper.

  1. The objective of the study could be more focused. There is no clear research question. This paper is rather a research summary than a research paper. I'm not saying that every paper has to use a confirmative approach, and it is not possible to write a paper without a clear hypothesis. Still, focused research objectives could help to write a more persuasive paper. The question about electoral turnout and fairness is a good direction.
  2. Independent variables: It is not explained why these particular variables were tested. Were these the ones that were found meaningful in previous papers? Or were these the ones that were included in all waves? This points back to the research question problem I had mentioned above. I had a feeling that the authors throw all the variables in a "pot" and then check which works. I also suggest adding an online appendix with the distribution of each variable that has been used in the paper.
  3. Selecting the relevant variables: I don't see why only those variables were reported that had consistent effects. I think that is particularly interesting if a variable has a mixed impact. The sample sizes are huge, so it could not cause any model fitting problems. 
  4. What regression model do the authors use? I assume a linear OLS regression. Please state this clearly. If they used linear regression, they could add BETA instead of B coeff. It could help to evaluate the strength of each variable. 
  5. The dependent variable is an ordinal variable. So it would be important to run ordered logistic regressions as a robustness check. 
  6. The sample sizes in the model description (table 3) are much smaller than what is reported in table 1. It seems that half of the sample disappeared at one point, and there is no clear explanation for this. I understand that some questions were not asked in the whole sample, but it might be essential to make this clear as possible. 
  7. I understand that party preference was not significant in the models. But this is not the same as the winner/loser effect. I don't think the particular party is interesting; rather, if they support that party who later won the election (or in the case of a referendum, the party who supported the winner referendum option). In some cases, it is pretty obvious who will win. In this case, if I know that my party will win, I might be optimistic about the upcoming fairness and – if I know that my party will lose, I might have concerns about the fairness of the elections. 
  8. The "Consequence of expected fairness" part is interesting, but it is hard to evaluate the results without seeing any models. I suggest adding least the models of the second analysis. Interaction terms are tricky to include in a logistic model (see the papers of Allison). The first and second analysis might be enough here.  
  9. Online data collection: BESIP in an online data collection that has some consequences on the generalizability of the results. People with higher political interest are overrepresented in this data. That is why so low the ratio of non-voters in the sample. I would like to see a paragraph about how this data collection method could affect the results. 

Minor ones.

  1. I suggest changing the direction of the dependent variable. The authors speak about fairness in the whole paper, and in the model, a higher value means unfairness.
  2. Character fonts are not identical throughout the whole paper. 
  3. 543 row: point missing at the end of the sentence. 

Round 2

Reviewer 1 Report

The authors of the manuscript have addressed my concerns and suggestions, and the revised version of the manuscript is clearer and more coherent. I also really liked that added Table 4 which enables a comparison between the two analyses. I do have one more comment, but I believe the authors can easily address it. This comment is intended to make the paper stronger and more coherent.

1) While the authors maintain in the revised manuscript that "which party one intends to vote for at the next General Election" is not (or not consistently) associated with prospective fairness expectation (page 10, lines 462-468), vote intention for the Conservative party seems (judging from Appendix C, page 29) to predict prospective fairness expectation rather consistently (significant coefficients in 4 out of 5 models). I would appreciate it if the authors elaborated on these findings. Moreover, it seems there is a mistake with regard to Conservative party voters in the 2017 General Election model, as the signs of this coefficient are negative in Table 4 (page 14) but positive in Appendix C. And I believe there is also a mistake in page 13, lines 563-570, where the authors are presenting the results among Conservative and Labour supporters (it should be the opposite "sign"…). Maybe I'm wrong on this, but I advise the author to double-check this.

2) A really minor comment: I believe that the short paragraph preceding Table 4 on page 13 (lines 581-585) should be dropped as it is redundant (similar text is shown in the table title and the table's note/legend).

Reviewer 2 Report

Thank you for the hard work the author(s) did to improve the paper based on the suggestions. I think the paper in the current form is much better than the previous one. The authors were able to answer the highlighted problems. Because of the mass amount of independent variables tested in the model, I still have some concerns about the results, but I can accept how they built up the analysis.

The authors’ comments about the standardized coeffs are correct; you can’t compare the effect size between models. But at least they allow the comparison within a model. But I don’t insist on using Beta in the presentation of tables.

I only have one more significant concern, the size of missing data in the models. As they now use listwise deletion, they assume the missing as MCAR. I don’t think it is true. I agree that it is hard to test the MAR assumption and even harder to find the relevant model, but the 50 percent missing rate is alarming. I’m not sure what software the authors use, but there are well-developed solutions to impute the data in R, even with multiple imputation techniques (I usually use the mice package in R). I would try this (or something close) to impute the independent variables. 

I appreciate the work the authors put into rewriting chapter 6. In its current form, I find it more convincing. Table 6 seems very interesting, although the low rate of non-voters decreases the validity of the results. Could you add the non-voters rate here for the sub-sample used in this analysis (without those who gave missing answers for any questions in the model). Is the “n” in this table the same for the baseline and the extended model? When they added the fairness belief attitude, I assumed some respondents could drop out who was in the baseline model. If the analysis sample is not the same, the AIC/BIC is biased. If the 'n' was the same, please ignore my comment. 
